# Survival Rates of Patients with Non-Small Cell Lung Cancer Depending on Lymph Node Metastasis: A Focus on Saliva

**DOI:** 10.3390/diagnostics11050912

**Published:** 2021-05-20

**Authors:** Lyudmila V. Bel’skaya, Elena A. Sarf, Victor K. Kosenok

**Affiliations:** 1Biochemistry Research Laboratory, Omsk State Pedagogical University, 644099 Omsk, Russia; nemcha@mail.ru; 2Department of Oncology, Omsk State Medical University, 644099 Omsk, Russia; victorkosenok@gmail.com

**Keywords:** non-small cell lung cancer, overall survival, histological type, lymph node metastases, treatment, saliva

## Abstract

The aim of this study was to compare overall survival (OS) rates at different pN stages of NSCLC depending on tumor characteristics and to assess the applicability of saliva biochemical markers as prognostic signs. The study included 239 patients with NSCLC (pN_0_-120, pN_1_-51, pN_2_-68). Saliva was analyzed for 34 biochemical indicators before the start of treatment. For pN_0_, the tumor size does not have a prognostic effect, but the histological type should be taken into account. For pN_1_ and pN_2_, long-term results are significantly worse in squamous cell cancer with a large tumor size. A larger volume of surgical treatment reduces the differences between OS. The statistically significant factors of an unfavorable prognosis at pN_0_ are the lactate dehydrogenase activity <1294 U/L and the level of diene conjugates >3.97 c.u. (HR = 3.48, 95% CI 1.21–9.85, *p* = 0.01541); at pN_1_, the content of imidazole compounds >0.296 mmol/L (HR = 6.75, 95% CI 1.28–34.57, *p* = 0.00822); at pN_2_ levels of protein <0.583 g/L and Schiff bases >0.602 c.u., as well as protein >0.583 g/L and Schiff bases <0.602 c.u. (HR = 2.07, 95% CI 1.47–8.93, *p* = 0.04351). Using salivary biochemical indicators, it is possible to carry out stratification into prognostic groups depending on the lymph node metastasis.

## 1. Introduction

Approximately 85% of lung cancers are non-small cell lung cancer (NSCLC). The most important parameters that determine treatment and survival in this group are the stage of the disease and metastases in the lymph nodes [1,2]. The degree of lymphogenous metastasis of NSCLC not only affects the prognosis of patients, but also largely determines the optimal treatment tactics [1]. Therefore, at stage pN_0–1_ the first and main stage of treatment is surgery, at pN_3_-chemotherapy and radiotherapy. The tactics of treating patients with pN_2_ have not yet been fully determined and are the subject of active discussion in the literature [3,4]. Recent practice guidelines consider chemotherapy and radiation therapy to treat patients with pN_2_, and do not recommend isolated or primary surgery [5]. Some supporters of reducing the volume of surgery consider it possible to apply individual schemes of lymph node dissection, focusing on the frequency of metastasis, the size and location of the tumor, the form of growth, and topography of the lymph nodes [6,7]. However, in patients with an early clinical stage of NSCLC, metastases in the lymph nodes are sometimes found during histopathological examination [8]. At the same time, even within one stage pN, the survival rate varies greatly depending on the size, histological type, degree of differentiation of the tumor and several other factors [9].

We have previously shown that several biochemical indicators of saliva can act as prognostic signs in NSCLC [10]. The aim of this study was to compare overall survival rates at different pN stages of NSCLC, depending on the characteristics of the tumor, and to assess the applicability of saliva biochemical indicators as prognostic signs.

## 2. Materials and Methods

### 2.1. Patient Population

The work is based on the results of examination and treatment of 320 patients (64 women, 256 men) admitted to the thoracic department of the Clinical Oncological Dispensary in Omsk in the period 2014–2017. Patients were enrolled after informed consent and the study was performed following the approval from the ethical committee of the Omsk Regional Clinical Oncological Dispensary (21 July 2016, Protocol No. 15) and in accordance with Helsinki principles.

Radical operations were performed in 191 patients (59.7%), including in the volume of lobectomy/bilobectomy (*n* = 163), pneumonectomy (*n* = 28). The combined treatment was required for 100 patients (31.3%). The indication for combined treatment, despite the radical nature of the operation, was the presence of metastases in the intrathoracic lymph nodes (N_1_, N_2_). As the second stage of treatment, external beam therapy was used up to a total focal dose of 46 Gy. Radiation treatment in an independent version was carried out in 43 patients (13.4%), 39 patients (12.2%) received chemotherapy, 19 patients (5.9%) received combined (radiation + chemotherapy). Based on the diagnostic results, special methods of treatment are not indicated in 28 patients (8.8%).

After histological verification, NSCLC was confirmed in all patients (133-squamous cell lung cancer, 187-adenocarcinoma). Additionally, we took into account the morphological forms of tumor growth: central (97 patients), peripheral (216 patients) and mediastinal (7 patients). Distant metastases were identified in 81 patients, so they were excluded from the study. The remaining 239 patients, depending on the lymph node metastases, were distributed as follows: pN_0_-120 (50.2%), pN_1_-51 (21.3%), pN_2_-68 (28.5%) patients. There were no gender and age differences between the groups of patients according to the status of lymph node metastases. The average age of the patients was 61.0 ± 1.34, 60.1 ± 2.04, and 59.1 ± 1.94 years for pN_0_, pN_1_, and pN2_,_ respectively.

### 2.2. Saliva Analysis

Saliva samples were collected at baseline, right before the start of treatment. Collection of saliva samples was carried out on an empty stomach after rinsing the mouth with water in the interval of 8–10 am by spitting into sterile polypropylene tubes, the salivation rate (mL/ min) was calculated. Saliva samples were centrifuged (10,000× *g* for 10 min) (CLb-16, Moscow, Russia), after which biochemical analysis was immediately performed without storage and freezing using the StatFax 3300 semi-automatic biochemical analyzer [11]. In all saliva samples, 34 biochemical parameters were determined, including pH, electrolyte levels, parameters of protein and lipid metabolism, and activity of metabolic and antioxidant enzymes as described previously [10].

### 2.3. Statistical Analysis

The total follow-up time was 6 years; the median follow-up time was 42 months. The patient’s overall survival (OS) was assessed from the date of hospitalization to the date of the last observation (censored) or the date of death of the patient (complete). OS was assessed using the Kaplan–Meier method with the presentation of survival curves and the calculation of the significance of differences by Log-rank (Statistica 10.0, StatSoft, Tulsa, OK, USA). Correction for uneven distribution according to the main initial criteria (gender, age, histological type, localization, tumor stage, treatment method) was performed using Cox regression. The description of the sample was made by calculating the median (Me) and interquartile range in the form of the 25th and 75th percentiles [LQ; UQ]. Differences were considered statistically significant at *p* < 0.05.

A univariate Cox proportional hazards regression analysis was initially variables carried out to investigate the relationships between salivary parameters and survival data. Finally, variables with *p* < 0.10 were chosen to formulate multivariate Cox proportional hazards regression models and determine the independent prognostic factors for OS. Hazard ratio (HR) was obtained with 95% confidence interval (CI). When evaluating the parameters of the regression model, those parameters for which the error was at least twice its standard error (*t* > 2.0) were considered statistically significant at the level of *p* < 0.05.

## 3. Results

### 3.1. Overall Survival Rates Depending on the Stage of the Disease, Histological Type, and Morphological Growth Forms of NSCLC

Median OS in the NSCLC group was 24.9 months. For patients without lymphogenous metastasis, the median OS was 36.1 months, with pN_1_ metastases, this value decreased to 18.2 months, and with pN_2_ to 14.3 months (Figure 1). The relative risk increases for stages pN_0_ vs. pN_1_ (HR = 5.85, 95% CI 2.71–12.31) and pN_0_ vs pN_2_ (HR = 10.55, 95% CI 4.73–22.80, *p* < 0.00001).

At the next stage of the study, subgroups were identified taking into account the characteristics of the tumor. Thus, with an increase in tumor size, OS naturally decrease, but the differences are statistically insignificant (Table 1). For pN_2_, the differences between the pT3 and pT4 stages are the smallest. When compared for one tumor size, for small tumors (T2) the relative risk increases statistically significantly between pN_0_ and pN_1_, while further changes are insignificant: HR = 7.22, 95% CI 2.63–19.24 (pN_0_ vs. pN_1_) and HR = 6.64, 95% CI 2.68–16.02 (pN_0_ vs. pN_2_). For T3, comparing pN_0_ vs. pN_1_, there is a 4-fold increase in risk (HR = 4.33, 95% CI 1.29–14.25), while comparing pN_0_ vs. pN_2_, there is a 12-fold increase in relative risk (HR = 11.82, 95% CI 2.23–10.48, *p* < 0.00001). For stage T4 with lesion of pN_1_, the minimum OS were revealed.

Taking into account the histological type of NSCLC, it was shown that regardless of the presence / absence and degree of lymph node involvement, the survival rates for squamous cell carcinoma are worse than for adenocarcinoma (Table 1). It should be noted that adenocarcinoma is more often detected at the pN_0_ stage (57.3 vs. 41.7%), while squamous cell carcinoma predominates at the pN_1_ and pN_2_ stages (16.8 vs. 26.9% and 25.9 vs. 31.5%, respectively). In general, patients with adenocarcinoma of the lung, even with lesions of the lymph nodes, have a more favorable prognosis than patients with squamous cell carcinoma (Table 1). For squamous cell lung cancer, the survival rate sharply decreases already at pN_1_, and then practically does not change. Apparently, it is for squamous cell lung cancer that metastatic lesions of the ipsilateral pulmonary, bronchopulmonary and/or lymph nodes of the lung root are a factor in the unfavorable prognosis of the disease.

In addition, we noted that regardless of the degree of damage to the lymph nodes, OS in central cancer is lower than in peripheral cancer (Table 1). This difference is most pronounced for pN_1_. Multiple lymph node lesions are characteristic of the mediastinal form of lung cancer; therefore, this subgroup is isolated only for pN_2_ and is characterized by a minimum OS (Table 1).

Depending on the differentiation of lung cancer, for pN_0_ the OS is significantly reduced only for undifferentiated cancer (Table 1), while for pN_1_ and pN_2_ it is already for the average degree of differentiation. For highly differentiated lung cancer, OS for pN_0_ and pN_1_ practically do not differ, while for pN_2_ they sharply decrease. Thus, the presence of even a single metastasis in the lymph nodes is an unfavorable prognosis factor in moderate and poorly differentiated lung tumors (Table 1). For highly differentiated lung cancer, the relative risk increases for pN_0_ vs. pN_1_ (HR = 3.83, 95% CI 0.77–18.85) and pN_0_ vs. pN_2_ (HR = 9.20, 95% CI 0.91–90.23), slightly increases for average differentiation for pN_0_ vs. pN_1_ (HR = 1.90, 95% CI 0.37–9.72) and increases statistically significantly for pN_0_ vs. pN_2_ (HR = 7.14, 95% CI 1.35–36.83). For poorly differentiated lung tumors, the risk increases for pN_0_ vs. pN_1_ (HR = 8.50, 95% CI 1.50–46.81) and pN_0_ vs. pN_2_ (HR = 12.28, 95% CI 3.23–45.10), for undifferentiated cancer we observe a similar trend: for pN_0_ vs. pN_1_ HR = 8.94 (95% CI 1.56–49.84) and pN_0_ vs. pN_2_ HR = 10.29 (95% CI 2.28–45.07, *p* < 0.00001).

### 3.2. The Predictive Value of the Type of Treatment 

For pN_0_, radical surgical and combined treatment is used; in the second case, OS is statistically significantly worse (Table 2). It is interesting to note that with lobectomy, including extended lobectomy, OS decreases with the transition from pN_0_ to pN_1_, but remains at the same level for pN_2_ (Table 2). For pneumonectomy, OS changes are not significant regardless of the degree of lymph node involvement (Table 2).

### 3.3. Predictive Value of Saliva Biochemical Indicators

By constructing a Cox regression model, we selected indicators that have a potential prognostic value in NSCLC at various pN stages (Figure 2). For pN_0_, such indicators include the activity of lactate dehydrogenase (LDH) and the level of diene conjugates (DC), for pN_1_—the level of imidazole compounds (ICs) and medium molecular weight toxins (MM), for pN_2_—the content of total protein and Schiff bases (SB) (Table 3).

When assessing the prognostic value of saliva biochemical indicators, the values of the median and interquartile range were used for the corresponding indicators in this group. At pN_0_, the LDH activity was 1294.0 [635.6; 1900.0] U/L, DC level was 3.97 [3.76; 4.19] c.u. Values of indicators LDH > 1294 U/L and DC < 3.97 c.u. are independent prognostically favorable signs (Table 3). For patients with salivary LDH activity before the start of treatment, more than 1294 U/L 1-, 3-, and 5-year OS were 91.4, 70.2 and 56.0%, less than 1294 U/L-89.7, 60.8 and 14.8% respectively. If the DC level is less than 3.97 c.u. OS was significantly higher than at a content of more than 3.97 c.u. (96.3, 74.5 and 41.4% vs. 85.2, 57.5 and 0%). With a favorable prognosis (LDH > 1294 U/L, DC < 3.97 c.u.) 1-, 3-, and 5-years OS was 96.5, 75.9, and 50.6%, whereas with an unfavorable (LDH < 1294 U/L, DC > 3.97 c.u.) OS was 84.8, 52.4 and 0%, respectively.

It should be noted that with an LDH activity of more than 1900 U/L, the relative risk was 3.8 times lower than with an activity of less than 636 U/L. The resulting value is statistically significant and can be used as an independent option (Table 3).

In the case of pN_1_, the content of imidazole compounds was ICs 0.296 [0.182; 0.455] mmol/L, middle molecular toxins MM-0.903 [0.832; 0.989] c.u. An independent prognostic sign is only the content of imidazole compounds (Table 3). For pN_1_, for patients with ICs content in saliva before the start of treatment less than 0.296 mmol/L 1-, 3-, and 5-year OS were 69.6, 38.1, and 16.7%, more than 0.296 mmol/L-53.8, 15.4, and 7.7%, respectively. If the MM level is less than 0.903 c.u. OS was slightly lower than at an MM content of more than 0.903 c.u. (60.0, 20.0 and 15.0% vs. 62.5, 32.4 and 10.8%). With a favorable prognosis (MM > 0.903 c.u., ICs < 0.296 mmol/L) for 1-, 3-, and 5-years, OS was 69.2, 44.9, and 16.7%, while with an unfavorable one, 53.3, 13.5, and 6.7%, respectively.

For pN_2_, the prognostic signs are protein content-0.583 [0.304; 1.044] g/L and the level of Schiff bases SB-0.602 [0.526; 0.669] c.u. Both indicators are not independent prognostic signs, therefore, combinations with unfavorable (Protein < 0.583 g/L and SB > 0.602 c.u.; Protein > 0.583 g/L and SB < 0.602 c.u.) and favorable prognosis (Protein > 0.583 g/L and SB > 0.602 c.u.; Protein < 0.583 g/L and SB < 0.602 c.u.) were considered (Table 3). For patients with a favorable prognosis, OS values were 65.5, 26.5, and 10.1%, with unfavorable ones-43.7, 10.1 and 0% for 1-, 3-, and 5-year survival rates, respectively.

We have presented the characteristics of cohorts depending on the differences in the biochemical composition of saliva with different status of lymph node involvement (Appendix A). It was shown that in addition to the difference in the biochemical composition of saliva, there are no other statistically significant differences between subgroups, including age, gender, pT, histological type, growth form, type of treatment, smoking, and relapse status). The only identified difference is the lower recurrence rate in the group of patients with a favorable prognosis for the biochemical composition of saliva for pN_0_ (Appendix A).

### 3.4. Multivariate Survival Analysis Using the Cox Regression Model

Multivariate analysis, including the stage of the disease (pT), histological structure, growth form, type of treatment, as well as the studied biochemical indicators for each pN group, showed that in all cases the biochemical parameters of saliva are independent factors in predicting the overall survival of patients with lung cancer (Table 4).

## 4. Discussion

Traditional tumor characteristics such as differentiation, tumor invasion, lymph node metastasis, and TNM (Tumor, Nodes, and Metastasis) stage classification are not the only aspects that determine the prognosis of the disease [12,13,14]. For prognostic purposes, the use of several groups of biomarkers is described. Therefore, the most significant is the study of genetic, epigenetic, proteomic, metabolic markers, as well as the profile of synthesis and the level of microRNA [15,16,17,18]. These markers are detected in tumor tissue, serum and blood plasma, and exhaled air [19,20].

In the literature, there are sporadic data on the study of the composition of saliva in lung cancer, including for prognostic purposes [21,22,23,24]. We have shown for the first time the fundamental possibility of using saliva biochemical indicators for predicting the course of lung cancer [10,25]. Of the indicators that were selected in regression analysis, only LDH was previously mentioned in the literature as a prognostic sign for blood plasma in lung cancer [26,27,28,29,30]. In this regard, comparison with literature data is not possible. It should also be noted that in previous studies we have shown that for most biochemical markers of saliva correlations with the composition of blood plasma are weak or absent altogether, therefore the values of biochemical markers of saliva should be considered to be independent and set their own criteria for norm and pathology [31].

The use of biochemical indicators of saliva allows obtaining prognostic data comparable to those for the characteristics of the tumor. In particular, in the absence of lymph node metastasis, tumor size is not a significant prognostic sign. However, the histological type of NSCLC is prognostically important; poor prognosis is associated with squamous cell lung cancer and undifferentiated cancer [32]. The statistically significant factors of unfavorable prognosis are LDH activity less than 1294 U/L and a DC level of more than 3.97 c.u. (Table 4).

With pN_1_, unfavorable prognosis factors include large tumor size (pT_4_), squamous histological type, central growth, as well as low differentiation or undifferentiated cancer (Table 1). An additional biochemical indicator in this case is the level of ICs. The concentration of ICs greater than 0.296 mmol/L is a statistically significant independent factor of poor prognosis (HR = 6.75, 95% CI 1.28–34.57, *p* = 0.00822). Multivariate analysis showed that the ICs level is the only independent prognostic factor for the group of patients with pN_1_ (Table 4).

At pN_2_, a poor prognosis is associated with squamous cell carcinoma, mediastinal tumor growth, and any differentiation other than highly differentiated tumors. According to our data, the size of the primary tumor does not statistically significantly affect the prognosis, which may be the result of an insufficient sample size. In the literature, age and pT stage are considered prognostically important [33]. Of the biochemical indicators, only the combination of indicators “Protein + SB” can be attributed to independent prognostic signs (Table 3 and Table 4). An unfavorable prognosis is typical for groups of patients with a protein content of less than 0.583 g/L and an SB of more than 0.602 c.u., as well as a protein level of more than 0.583 g/L and an SB of less than 0.602 c.u. (HR = 2.07, 95% CI 1.47–8.93, *p* = 0.04351).

In all cases, the type of treatment is a significant prognostic factor, which is quite natural. It is interesting to note that with an increase in the volume of surgery, the differences in OS medians decrease (Table 2). Thus, in the case of pneumonectomy, the median OS is 18.5, 17.8, and 17.1 months for stages pN_0_, pN_1_, and pN_2_, respectively.

An interesting result of our study is that with different degrees of damage to the lymph nodes, different biochemical parameters of saliva are used as prognostically important parameters. It can be assumed that this is due to the depth of the metabolic changes occurring in lung cancer. Therefore, in the absence of damage to the lymph nodes, the prognosis is determined by the activity of LDH as the main metabolic enzyme, the activity of which changes in many types of cancer, including lung cancer. The level of diene conjugates, which characterize the content of primary lipid peroxidation products, is also prognostically important. At pN_1_, the predictive factors include the total content of protein toxins and imidazole compounds, while at pN_2_ toxic Schiff bases, which are the end products of lipid peroxidation, are prognostically important. However, this hypothesis requires additional verification in the course of further research.

The limitations of the study are related to the fact that it was not assessed whether the lesions of the pN_1_ and pN_2_ lymph nodes are single or multiple [34]. The limitations should also include the small sample size, which reduces the statistical significance of the data obtained and limits the possibility of dividing into subgroups. Further studies are warranted to confirm our observation.

## 5. Conclusions

In the absence of metastases in regional lymph nodes, the size of the primary tumor has no significant prognostic effect; however, the histological type of tumor should be taken into account. For stages pN_1_ and pN_2_, long-term results are significantly worse with a large tumor size, and the presence of histology of squamous cell lung cancer critically decreases the median OS in these groups. The degree of tumor differentiation at pN_0_ has practically no effect on OS, whereas for all types of tumors, except for highly differentiated ones, the median OS sharply decreases at pN_1_ and pN_2_. A larger volume of surgical treatment reduces the differences between OS in the study groups. It has been shown for the first time that using biochemical indicators of saliva, additional stratification into prognostic groups can be carried out, depending on the presence / absence and the prevalence of regional metastasis.

## Figures and Tables

**Figure 1 diagnostics-11-00912-f001:**
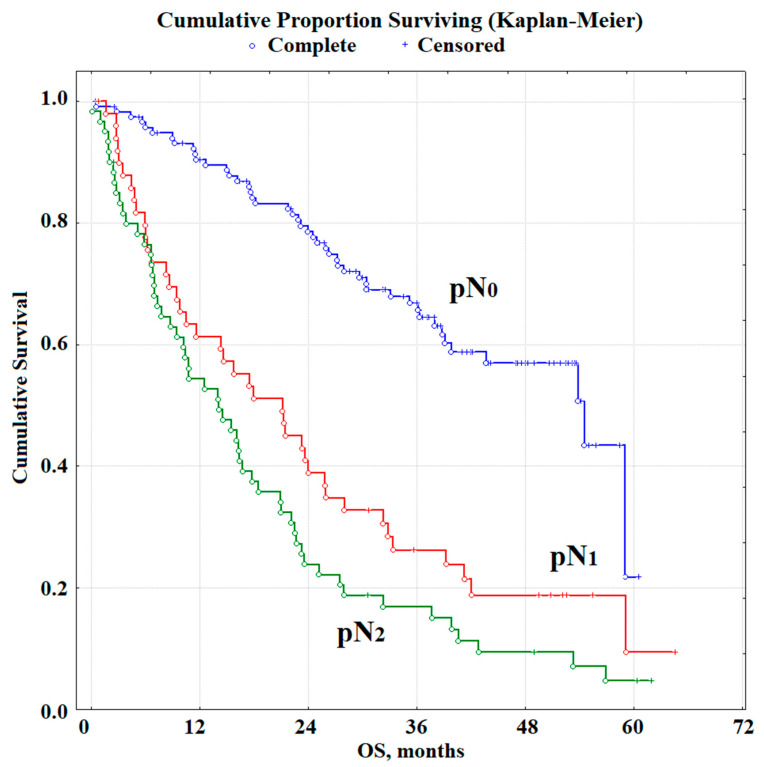
Kaplan–Meier survival curves illustrating the impact of pN stage on OS (*p* < 0.00001).

**Figure 2 diagnostics-11-00912-f002:**
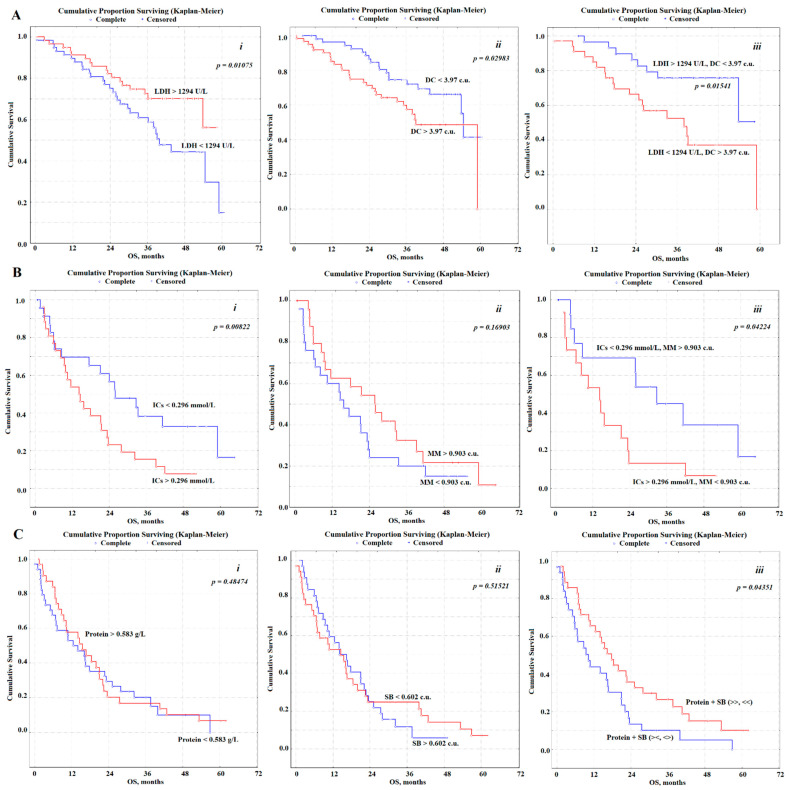
Kaplan–Meier survival curves illustrating the impact of biochemical salivary markers on OS. (**A**): N_0_—LDH (*i*), DC (*ii*) and LDH + DK (*iii*); (**B**): N_1_—ICs (*i*), MM (*ii*) and ICs + MM (*iii*); (**C**): N_2_—Protein (*i*), SB (*ii*) and Protein + SB (*iii*). LDH—lactate dehydrogenase, DC—diene conjugates, ICs—imidazole compounds, MM—middle molecular toxins, SB—Schiff bases.

**Table 1 diagnostics-11-00912-t001:** Overall survival rates depending on tumor size, histological type, growth form, and degree of differentiation of lung cancer.

LNS	Variables	HR (95% CI)	*p*-Value	OS, Months
**pT**				
**pN_0_**	pT_1_, *n* = 18	1	0.03363	40.4
	pT_2_, *n* = 80	1.31 (0.50–3.43)		36.8
	pT_3_, *n* = 22	2.18 (0.67–7.03)		28.9
**pN_1_**	pT_2_, *n* = 20	1	0.00223	29.6
	pT_3_, *n* = 23	1.10 (0.31–3.93)		19.7
	pT_4_, *n* = 8	2.40 (0.25–22.75)		5.9
**pN_2_**	pT_2_, *n* = 25	1	0.18914	16.8
	pT_3_, *n* = 18	2.96 (0.57–15.20)		10.2
	pT_4_, *n* = 25	3.85 (0.94–15.56)		11.0
**Histological Subtype**				
**pN_0_**	ADC, *n* = 75	1	0.03366	36.5
	SCC, *n* = 45	2.75 (1.28–5.86) *		32.7
**pN_1_**	ADC, *n* = 22	1	0.03069	26.3
	SCC, *n* = 29	2.13 (0.30–4.27)		9.6
**pN_2_**	ADC, *n* = 34	1	0.07122	16.6
	SCC, *n* = 34	2.21 (0.51–9.55)		9.7
**Morphological Growth Forms**				
**pN_0_**	Peripheral, *n* = 106 ADC-71, SCC-35	1	0.80773	36.7
	Central, *n* = 14 ADC-3, SCC-11	1.29 (0.46–3.59)		31.2
**pN_1_**	Peripheral, *n* = 28 ADC-19, SCC-9	1	0.08248	25.3
	Central, *n* = 23 ADC-3, SCC-20	1.26 (0.36–4.34)		13.5
**pN_2_**	Peripheral, *n* = 30 ADC-22, SCC-8	1	0.20789	16.0
	Central, *n* = 31 ADC-10, SCC-21	2.22 (0.62–7.90)		10.8
	Mediastinal, *n* = 7 ADC-0, SCC-7	2.89 (0.33–24.84)		9.8
**Degree of Differentiation**				
**pN_0_**	G1, *n* = 30	1	0.06772	38.7
	G2, *n* = 38	1.61 (0.59–4.37)		36.7
	G3, *n* = 27	1.22 (0.41–3.62)		32.3
	G4, *n* = 25	1.42 (0.45–4.43)		22.5
**pN_1_**	G1, *n* = 10	1	0.06314	37.9
	G2, *n* = 13	0.80 (0.10–6.27)		16.0
	G3, *n* = 8	2.70 (0.33–21.52)		18.2
	G4, *n* = 20	3.30 (0.42–25.79)		11.8
**pN_2_**	G1, *n* = 12	1	0.47036	25.2
	G2, *n* = 18	1.25 (0.09–17.63)		9.7
	G3, *n* = 9	1.63 (0.14–18.12)		11.8
	G4, *n* = 29	1.58 (0.13–19.04)		10.1

Note. *—differences are statistically significant, *p* < 0.05; ADC—adenocarcinoma, SCC—squamous cell carcinoma; G1—highly, G2—moderately, and G3—poorly differentiated, G4—undifferentiated lung cancer. LNS—lymph node status, OS—overall survival.

**Table 2 diagnostics-11-00912-t002:** Overall survival rates depending on the type of treatment and the volume of surgery.

LNS	Variables	HR (95% CI)	*p*-Value	OS, Months
**Treatment Type**				
**pN_0_**	Radical, *n* = 79	1	0.00196	37.7
	Combined, *n* = 36	2.66 (1.22–5.71) *		30.9
**pN_1_**	Combined, *n* = 27	1	0.00738	26.3
	Palliative, *n* = 19	10.91 (1.29–89.53) *		11.2
**pN_2_**	Combined, *n* = 19	1	0.08023	23.9
	Palliative, *n* = 36	2.63 (0.76–9.04)		12.7
**The Extent of Surgical Treatment**				
**pN_0_**	Lobectomy, *n* = 75	1	0.00021	37.6
	Bilobectomy, *n* = 33	1.11 (0.49–2.55)		31.2
	Pneumonectomy, *n* = 7	3.87 (0.86–17.10)		18.5
	No surgery, *n* = 5	6.96 (1.31–36.00) *		13.4
**pN_1_**	Lobectomy, *n* = 14	1	0.05689	32.8
	Bilobectomy, *n* = 7	0.62 (0.11–3.63)		21.8
	Pneumonectomy, *n* = 9	0.25 (0.04–1.38)		17.8
	No surgery, *n* = 21	8.31 (0.84–79.56)		11.2
**pN_2_**	Lobectomy, *n* = 4	1	0.56381	22.6
	Bilobectomy, *n* = 8	2.00 (0.14–27.78)		20.5
	Pneumonectomy, *n* = 12	4.33 (0.33–56.10)		17.1
	No surgery, *n* = 44	2.63 (0.45–15.00)		11.0

Note. LNS—lymph node status, OS—overall survival, *—differences are statistically significant, *p* < 0.05.

**Table 3 diagnostics-11-00912-t003:** Prognostic value of saliva biochemical markers depending on the prevalence of metastases in the lymph nodes.

Indicators	Variables	HR (95% CI)	*p*-Value	OS, Months
**pN_0_**				
**LDH, U/L**	<1294, *n* = 60	1	0.01075	30.6
	>1294, *n* = 60	0.42 (0.20–0.90) *		36.6
**LDH, U/L**	<636, *n* = 46	1	0.16263	30.1
	636–1900, *n* = 47	0.47 (0.21–1.09)		37.2
	>1900, *n* = 27	0.26 (0.09–0.77) *		36.4
**DC, c.u.**	<3.97, *n* = 56	1	0.02983	38.2
	>3.97, *n* = 64	1.64 (0.78–3.44)		30.3
**DC, c.u.**	<3.76, *n* = 29	1	0.14553	41.3
	3.76–4.19, *n* = 70	1.48 (0.59–3.69)		36.5
	>4.16, *n* = 21	1.67 (0.52–5.30)		23.6
**LDH + DC**	>1294, <3.97, *n* = 30	1	0.01541	39.6
	<1294, >3.97, *n* = 34	3.48 (1.21–9.85) *		28.7
**pN_1_**				
**ICs, mmol/L**	<0.296, *n* = 25	1	0.00822	26.2
	>0.296, *n* = 26	6.75 (1.28–34.57) *		14.7
**ICs, mmol/L**	<0.182, *n* = 13	1	0.00639	26.2
	0.182–0.455, *n* = 25	0.64 (0.14–2.95)		23.7
	>0.455, *n* = 13	4.20 (1.38–45.31) *		6.8
**MM, c.u.**	<0.903, *n* = 25	1	0.16903	16.0
	<0.903, *n* = 26	0.52 (0.13–2.04)		24.0
**MM, c.u.**	<0.832, *n* = 13	1	0.10240	14.9
	0.832–0.989, *n* = 26	0.28 (0.03–2.58)		18.8
	>0.989, *n* = 12	0.17 (0.02–0.97) *		28.7
**ICs + MM**	<0.296, >0.903, *n* = 15	1	0.04224	26.3
	>0.296, <0.903, *n* = 15	9.80 (1.04–89.91) *		14.5
**pN_2_**				
**Protein, g/L**	<0.583, *n* = 34	1	0.48474	13.5
	>0.583, *n* = 33	0.75 (0.18–3.04)		14.3
**SB, c.u.**	<0.602, *n* = 34	1	0.51521	14.3
	>0.602, *n* = 34	1.29 (0.32–5.24)		12.9
**Protein + SB**	><, <>, *n* = 33	1	0.04351	9.3
	>>, <<, *n* = 35	0.50 (0.12–0.98) *		16.6

Note. *—differences are statistically significant, *p* < 0.05; LDH—lactate dehydrogenase, DC—diene conjugates, ICs—imidazole compounds, MM—middle molecular toxins, SB—Schiff bases.

**Table 4 diagnostics-11-00912-t004:** Results of multivariate survival analysis using the Cox regression model.

Prognostic Factors	β	Standard Error	*t*-Value	*p*-Value
**N_0_** (**χ^2^ = 34.55, *p* < 0.00001**)				
**pT**	0.2020	0.2938	0.6875	0.4918
**Histological Subtype**	0.8011	0.3154	2.540	**0.0111**
**Morphological Growth Forms**	−1.031	0.5382	−1.915	0.0555
**Treatment Types**	1.220	0.2509	4.862	**0.0000**
**LDH**	−1.094	0.3440	−3.179	**0.0015**
**DC**	0.6192	0.3083	2.008	**0.0447**
**N_1_** (**χ^2^ = 20.57, *p* = 0.00446**)				
**pT**	0.6598	0.3417	1.931	0.0535
**Histological Subtype**	0.6680	0.4306	1.551	0.1208
**Morphological Growth Forms**	−0.8718	0.4470	−1.951	0.0511
**Treatment Types**	0.8776	0.4488	1.956	0.0505
**ICs**	0.6709	0.3342	2.007	**0.0448**
**MM**	−0.2486	0.3269	−0.7606	0.4469
**N_2_** (**χ^2^ = 18.80, *p* = 0.00884**)				
**pT**	−0.1463	0.2107	−0.6943	0.4875
**Histological Subtype**	0.4886	0.3275	1.492	0.1357
**Morphological Growth Forms**	0.2740	0.2860	0.9582	0.3380
**Treatment Types**	0.6314	0.2707	2.333	**0.0197**
**Protein**	−0.0891	0.2878	−0.3095	0.7569
**SB**	0.0806	0.2869	0.2808	0.7789
**Protein + SB**	−0.6886	0.3007	−2.290	**0.0220**

Note. Histological subtype (ADC = 0, SCC = 1); Morphological growth forms (Peripheral = 0, Central = 1); Treatment Types (Radical = 0, Combined = 1, Palliative = 2); LDH, DC, ICs, MM, Protein and SB (continue variables); Protein + SB (Favorable prognosis = 0, unfavorable prognosis = 1). Statistical results with *p* < 0.05 are bolded. LDH—lactate dehydrogenase, DC—diene conjugates, ICs—imidazole compounds, MM—middle molecular toxins, SB—Schiff bases.

## Data Availability

The data presented in this study are available on request from the corresponding author. The data are not publicly available due to they are required for the preparation of a PhD thesis.

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
