# Peer review of "Survival Rates of Patients with Non-Small Cell Lung Cancer Depending on Lymph Node Metastasis: A Focus on Saliva"

_diagnostics, 2021, doi:10.3390/diagnostics11050912_

Round 1
Reviewer 1 Report
This manuscript is a continuation of previous papers of these authors, which was published in the Diagnostics 2020, 10(4), 186 and Biomedical Chemistry: Research and Methods, 3(3), e00133. It was shown there, that imidazole compounds (ICs) and salivary lactate dehydrogenase activity can be significant independent factors in the poor prognosis of lung cancer. In this article the authors have showed that salivary biochemical indicators can be useful for patients’ stratification into prognostic groups depending on the lymph node metastasis. Although using chemical composition of saliva seems to be a very attractive prognostic marker in this type of cancer, I have some doubts whether these results contribute something more than the results presented in their previous articles. My concerns are:
- The Kaplan Meier survival curves illustrating the impact of biochemical salivary markers on OS were showed without p-values, although the authors declared their testing using log-rank test in “Material and methods” section. Except LDH measurements in N0 patients, the Kaplan Meier curves seem similar, therefore the prognostic usefulness of the rest of analyzed compounds is not obvious.
- Can the authors provide the characteristic (age, histological subtype, pT, treatment types, morphological growth forms, comorbidities) of the cohorts with significantly different OS in terms of the chemical composition of saliva (pN0- LDH, LDH+DC; pN1 – ICs, MM, ICs+MM; pN2 - Protein + SB)? This could allow for a real assessment of the prognostic value of these parameters on the overall survival.
- Why different chemical compounds in saliva are related with the prognosis in patients classified as pN1 and pN2? What could be the biological explanation for that? It should be discussed in discussion section.
- There is some data that smoking can modulate chemical and microbiological composition of saliva. Have the authors taken this into account in their analysis?
- The age structure for the whole analyzed cohort should be provided
Author Response
Responses to reviewer’s comments
The authors of the article are grateful to the reviewers for valuable comments that made it possible to eliminate inaccuracies and make the article better.
Reviewer 1
This manuscript is a continuation of previous papers of these authors, which was published in the Diagnostics 2020, 10(4), 186 and Biomedical Chemistry: Research and Methods, 3(3), e00133. It was shown there, that imidazole compounds (ICs) and salivary lactate dehydrogenase activity can be significant independent factors in the poor prognosis of lung cancer. In this article the authors have showed that salivary biochemical indicators can be useful for patients’ stratification into prognostic groups depending on the lymph node metastasis. Although using chemical composition of saliva seems to be a very attractive prognostic marker in this type of cancer, I have some doubts whether these results contribute something more than the results presented in their previous articles.
1. The Kaplan Meier survival curves illustrating the impact of biochemical salivary markers on OS were showed without p-values, although the authors declared their testing using log-rank test in “Material and methods” section. Except LDH measurements in N0 patients, the Kaplan Meier curves seem similar, therefore the prognostic usefulness of the rest of analyzed compounds is not obvious.
The p values are given in Table 3 for each case, however, for convenience we have given the corresponding values in the graphs (Figure 2).
2. Can the authors provide the characteristic (age, histological subtype, pT, treatment types, morphological growth forms, comorbidities) of the cohorts with significantly different OS in terms of the chemical composition of saliva (pN0- LDH, LDH+DC; pN1 – ICs, MM, ICs+MM; pN2 - Protein + SB)? This could allow for a real assessment of the prognostic value of these parameters on the overall survival.
We have presented the characteristics of cohorts depending on the differences in the biochemical composition of saliva with different status of lymph node involvement (Supplementary materials, Tables S1-3). It was shown that in addition to the difference in the biochemical composition of saliva, there are no other statistically significant differences between subgroups, including age, sex, pT, histological type, growth form, type of treatment, etc.). The only identified difference is the lower recurrence rate in the group of patients with a favorable prognosis for the biochemical composition of saliva for pN0.
3. Why different chemical compounds in saliva are related with the prognosis in patients classified as pN1 and pN2? What could be the biological explanation for that? It should be discussed in discussion section.
An interesting result of our study is that with different degrees of damage to the lymph nodes, different biochemical parameters of saliva are used as prognostically important parameters. It can be assumed that this is due to the depth of the metabolic changes occurring in lung cancer. So, in the absence of damage to the lymph nodes, the prognosis is determined by the activity of LDH as the main metabolic enzyme, the activity of which changes in many types of cancer, including lung cancer. The level of diene conjugates, which characterize the content of primary lipid peroxidation products, is also prognostically important. At pN1, the predictive factors include the total content of protein toxins and imidazole compounds, while at pN2 toxic Schiff bases, which are the end products of lipid peroxidation, are prognostically important. However, this hypothesis requires additional verification in the course of further research.
4. There is some data that smoking can modulate chemical and microbiological composition of saliva. Have the authors taken this into account in their analysis?
Yes, we took into account the status of smoking, but the distribution of smokers and non-smokers in the groups is approximately the same and does not significantly affect the study result.
5. The age structure for the whole analyzed cohort should be provided
There were no gender and age differences between the groups of patients according to the status of lymph node metastases. The average age of the patients was 61.0±1.34, 60.1±2.04, and 59.1±1.94 years for pN0, pN1 and pN2 respectively. In all cases, when dividing into subgroups, a check was carried out and age was not a statistically significant differentiating factor, therefore, age data are not given in order not to overload the tables.

Reviewer 2 Report
Survival rates of patients with non-small cell lung cancer depending on lymph node metastasis: A focus on saliva by Bel’skaya et al is interesting however the study needs additional analysis or experimental proof to strengthen the manuscript to be published in Diagnostics
Strength of the manuscript: The study is on clinical samples and translational with 239 patients samples.
Weakness of the study and major comments: The aim of this study was to compare overall survival (OS) rates at different pN stages of 8 NSCLC depending on tumor characteristics and to assess the applicability of saliva biochemical markers as prognostic signs. The study included 239 patients with NSCLC (pN0 – 120, pN1 – 51, pN2 10 – 68). However the authors did the extensive analysis on 239 patient samples. The major concern of the manuscript is that the authors show only observational data including frequency of metastasis, size and location of the tumor, the form of growth and topography of the lymph nodes and no biological data or readout
for the NSCLC metastases. If the authors could show the peripheral blood serum data that supports tumor size, growth rate and stage of the NSCLC will help to improve the manuscripts quality.
It requires a moderate english editing
Author Response
Responses to reviewer’s comments
The authors of the article are grateful to the reviewers for valuable comments that made it possible to eliminate inaccuracies and make the article better.
Reviewer 2
Survival rates of patients with non-small cell lung cancer depending on lymph node metastasis: A focus on saliva by Bel’skaya et al is interesting however the study needs additional analysis or experimental proof to strengthen the manuscript to be published in Diagnostics
Strength of the manuscript: The study is on clinical samples and translational with 239 patients samples.
Weakness of the study and major comments: The aim of this study was to compare overall survival (OS) rates at different pN stages of 8 NSCLC depending on tumor characteristics and to assess the applicability of saliva biochemical markers as prognostic signs. The study included 239 patients with NSCLC (pN0 – 120, pN1 – 51, pN2 10 – 68). However the authors did the extensive analysis on 239 patient samples. The major concern of the manuscript is that the authors show only observational data including frequency of metastasis, size and location of the tumor, the form of growth and topography of the lymph nodes and no biological data or readout for the NSCLC metastases. If the authors could show the peripheral blood serum data that supports tumor size, growth rate and stage of the NSCLC will help to improve the manuscripts quality.
The objectives of this study were to analyze the biochemical analysis of saliva in patients with lung cancer, depending on the defeat of the lymph nodes. However, we did not collect biological data in more detail; we took into account only the pN staging performed by the oncologist. Tumor size, growth shape, and stage of NSCLC were determined by computed tomography and histological verification. We did not use any diagnostic options other than the standard procedure performed upon admission to the hospital. Peripheral blood serum data are not used at our cancer center to confirm tumor size and other characteristics.

Round 2
Reviewer 1 Report
I am comfortable with the modifications made by the authors and I have no further comments.